# Effect of Polytetrafluoroethylene (PTFE) Content on the Properties of Ni-Cu-P-PTFE Composite Coatings

**DOI:** 10.3390/ma16051966

**Published:** 2023-02-27

**Authors:** Xinghua Liang, Penggui Wu, Lingxiao Lan, Yujiang Wang, Yujuan Ning, Yu Wang, Yunmei Qin

**Affiliations:** 1Guangxi Key Laboratory of Automobile Components and Vehicle Technology, Guangxi University of Science and Technology, Liuzhou 545006, China; 2Guangdong Provincial Key Laboratory of Modern Surface Engineering Technology, Institute of New Materials, Guangdong Academy of Sciences, Guangzhou 510651, China

**Keywords:** Q235B mild steel, polytetrafluoroethylene, chemically plated, composite coating, corrosion resistance

## Abstract

Q235B mild steel has the advantages of good mechanical properties, welding properties, and low cost, and it is widely used in bridges, energy fields, and marine equipment. However, Q235B low-carbon steel is prone to serious pitting corrosion in urban water and sea water with high chloride ions (Cl^−^), which restricts its application and development. Herein, to explore the effects of different concentrations of polytetrafluoroethylene (PTFE) on the physical phase composition, the properties of Ni-Cu-P-PTFE composite coatings were studied. The Ni-Cu-P-PTFE composite coatings with PTFE concentrations of 10 mL/L, 15 mL/L, and 20 mL/L were prepared on the surface of Q235B mild steel by the chemical composite plating method. The surface morphology, elemental content distribution, phase composition, surface roughness, Vickers hardness, corrosion current density, and corrosion potential of the composite coatings were analyzed by scanning electron microscopy (SEM), energy dispersive spectrometry (EDS), X-ray diffraction (XRD), three-dimensional profile, Vickers hardness, electrochemical impedance spectroscopy (EIS), and Tafel curve test methods. The electrochemical corrosion results showed that the corrosion current density of the composite coating with a PTFE concentration of 10 mL/L in 3.5 wt% NaCl solution was 7.255 × 10^−6^ A∙cm^−2^, and the corrosion voltage was −0.314 V. The corrosion current density of the 10 mL/L composite plating was the lowest, the corrosion voltage positive shift was the highest, and the EIS arc diameter of the 10 mL/L composite plating was also the largest, which indicated that the 10 mL/L composite plating had the best corrosion resistance. Ni-Cu-P-PTFE composite coating significantly enhanced the corrosion resistance of Q235B mild steel in 3.5 wt% NaCl solution. This work provides a feasible strategy for an anti-corrosion design of Q235B mild steel.

## 1. Introduction

Q235B mild steel with green, environmentally friendly, and cost-effective advantages not only has good mechanical properties and welding properties but also corrosion resistance under environments with carbon dioxide (CO_2_) and hydrogen sulfide (H_2_S). It is widely used in construction machinery, weaponry, resource environments, energy power, and marine applications [1,2,3,4]. However, Q235B mild steel is prone to severe pitting corrosion, such as the formation of water and seawater with a high chloride ion (Cl^−^) content, in working environments, thus limiting the development of Q235B mild steel applications [5,6,7]. It is well known that chemically deposited nickel–phosphorus (Ni-P) coatings are widely used for surface corrosion protection of various mild steels due to their unique properties, such as corrosion resistance, wear resistance, paramagnetic properties, hardness, and electrocatalytic activity of hydrogen precipitation. The performance of Ni-P alloy coatings can be effectively improved by combining nanoparticles with Ni-P coatings through chemical co-deposition techniques to form functional nanocomposite coatings [8,9,10]. Various nanoparticles can be used to enhance the performance of Ni-P alloy coatings, such as hard particles, including silicon carbide (SiC), tungsten carbide (WC), alumina (Al_2_O_3_), zinc oxide (ZnO), titanium oxide (TiO_2_); metals, such as tungsten (W), iron (Fe), and copper (Cu); and lubricant particles, including graphite and polytetrafluoroethylene (PTFE) [11,12,13,14].

At present, most studies on chemical nickel plating are limited to binary and ternary alloy plating, while few studies have reported on the incorporation of polymeric compounds in binary and ternary alloy plating [13,15,16]. Polytetrafluoroethylene (PTFE), commonly known as the “plastic king”, is a polymer compound made of tetrafluoroethylene by polymerization that has excellent non-stick properties, wear resistance, anti-bonding, high dry lubricity, a low friction coefficient, a high melting point, low surface energy, a low friction coefficient, and other characteristics [17,18,19,20]; thus, its wear resistance and corrosion resistance have received the attention of the majority of researchers. Wan et al. [21] incorporated PTFE particles into Ni-B plating by chemical deposition, and the results of the study showed that the addition of PTFE particles led to a reduction in the friction coefficient and an improvement in the corrosion resistance of nickel–boron (Ni-B) plating. Zhou Yan et al. [22] prepared nickel–phosphorus–polytetrafluoroethylene (Ni-P-PTFE) composite plating by chemical nickel plating on the surface of landscape steel structures and explored the effect of different PTFE concentrations on the corrosion resistance of Ni-P-PTFE composite plating, and the results showed that Ni-P-PTFE composite plating with low PTFE concentrations had better corrosion resistance. Wang et al. [23] prepared Ni-P-PTFE composite plating by electrochemical deposition on the surface of Ni-P-PTFE composite coatings with different PTFE concentrations, which were prepared by electrochemical deposition on the surface of low-carbon steel. The Ni-P-PTFE composite coatings were placed in bacterial and sterile seawater for corrosion tests, and the results of the study showed that Ni-P-PTFE composite coatings with trace amounts of PTFE added had excellent corrosion resistance. Cheng et al. [24] investigated the effect of PTFE and surfactant concentrations on the deposition rate and surface free energy of nickel–copper–phosphorus–polytetrafluoroethylene (Ni-Cu-P-PTFE) composite plating, and the experimental results showed that the deposition rate of the Ni-Cu-P-PTFE composite plating showed a trend of increasing and then decreasing with the increase in PTFE and surfactant concentrations. In addition, the surface free energy of the Ni-Cu-P-PTFE composite plating decreased with the increase of PTFE concentration. The surface free energy of the P-PTFE composite coating decreased with the increase in PTFE concentration. The anti-fouling experimental results also showed that the Ni-Cu-P-PTFE composite coating had better anti-fouling properties than mild steel without composite coating. Mafi et al. [25] chemically deposited Ni-P-PTFE composite coating on a mild steel surface using different surfactants, and it was found that after adding cetyltrimethylammonium bromide (CTAB) and physical vapor deposition (PVP) to the Ni-P-PTFE composite coating, PTFE could be well dispersed and adsorbed on the substrate surface uniformly, and the corrosion resistance of the Ni-P-PTFE composite coating was significantly improved. When the concentration of CTAB was 0.3 g/L, the corrosion resistance of the Ni-P-PTFE composite coating was 16 times higher than that of the Ni-P sample. Liu et al. [26] prepared Ni-Cu-P-PTFE composite coating on the surface of 1020 mild steel and investigated the effect of copper ions (Cu^2+^), cationic surfactant, and PTFE concentrations in the plating solution and the temperature of the plating solution on the adsorption deposition rate of the Ni-Cu-P-PTFE composite coating. The results of the corrosion experiments with NaCl solution showed that the corrosion resistance of the Ni-Cu-P-PTFE composite coating was better than that of copper, 1020 mild steel, and stainless steel.

Therefore, the addition of PTFE to alloy plating can improve the corrosion resistance of the alloy plating. In addition, less research has reported on the chemical deposition of Ni-Cu-P-PTFE composite coatings; specifically, the effect of PTFE addition and the amount of PTFE concentration in the Ni-Cu-P-PTFE plating solution on Ni-Cu-P-PTFE composite coatings, physical phase composition, microscopic morphology, surface roughness, microhardness, and corrosion resistance is unclear.

Compared with other methods, such as electroplating [27,28] and electrodeposition [29], electroless deposition of composite coatings is considered a convenient and effective way to improve various physicochemical and operational properties of coatings [30]. Therefore, in this paper, a Ni-Cu-P-PTFE composite layer was prepared on a Q235B mild steel surface by adding different concentrations of PTFE to the plating solution using chemical plating technology, and the effect of the PTFE concentration on the physical composition, microscopic morphology, and the properties of the Ni-Cu-P-PTFE composite layer was explored to determine the optimal PEFE concentration, which further provides a theoretical basis for future steel surface treatment technology. A theoretical basis for future steel surface treatment technology is provided.

## 2. Experiment

### 2.1. Preparation and Process of Coating

Q235B mild steel (substrate size: Ø25.4 × 6 mm; composition (mass fraction): carbon (C) ≤ 0.12~0.22%, silicon (Si) ≤ 0.3%, manganese (Mn) ≤ 0.3%~0.7%, sulfur (S) ≤ 0.045%, phosphorus (P) ≤ 0.045%, chromium (Cr) ≤ 0.3%, nickel (Ni) ≤ 0.3%, copper (Cu) ≤ 0.3%; allowance: iron (Fe) was used as the base material for chemical plating. The main reagents used in chemical plating solutions, NiSO_4_∙6H_2_O (AR, 98.5%), CuSO_4_∙5H_2_O (AR), NaH_2_PO_2_∙H_2_O (AR), C_6_H_5_Na_3_O_7_∙2H_2_O (AR), C_3_H_6_O_3_ (AR, 85–90%), CTAB (99%), 60 wt% PTFE, NaOH (95%), Na_3_PO_4_ (AR), Na_2_CO_3_ (AR), C_34_H_62_O_11_ (AR), and H_2_SO_4_ (AR, 98%), were all from Maclean’s in Shanghai, China. CH_3_COONa (AR, 98%), C_2_H_5_NO_2_ (AR, 99.5–100.5%), and CH_4_N_2_S (AR, 99%) were from RHAWN’s in Shanghai, China.

Chemical deposition was carried out on the surface of Q235B mild steel (Ø25.4 × 6 mm). The substrate was pretreated simply before chemical deposition. The specific electroless plating experiment process was as follows. First, the substrate was polished by 400 #, 800 #, 1200 #, 1600 #, 2000 #, and 2400 # sandpaper. Second, the substrate was put in a solution of 25 g/L NaOH + 35 g/L Na_3_PO_4_ + 25 g/L Na_2_CO_3_ + 5 mL/L C_34_H_62_O_11_ heated to 85 °C for 15 min to remove oil. Using deionized water, the surface of the substrate was cleaned to remove the oil. Surface derusting was performed in 10% (volume fraction) H_2_SO_4_ pickling at room temperature for 1–2 min. Subsequently, the substrate was neutralized with 5% (volume fraction) NaOH for 15 s before repeated rinsing and drying with deionized water. In 5% (volume fraction) sulfuric acid, it was activated at room temperature and then repeatedly washed with deionized water. Finally, the pretreated substrate was heated in an 85 °C blast dryer for 2 min and then quickly placed in the plating solution at the set temperature for chemical deposition for 2 h. After the end of the experiment, the sample was removed, cleaned, and blow-dried. The flow chart of the experimental process setup is shown in Figure 1.

The reagents used in the experiment and the composition and process conditions of the bath for Ni-Cu-P-PTFE composite coatings are shown in Table 1.

### 2.2. Coating Structure Characterization and Phase Analysis

The surface, cross-section morphology, and cross-section element distribution of the composite coatings were observed and analyzed by scanning electron microscopy (SEM, ZEISS-EV0-MA15, Hitachi High-Tech, Tokyo, Japan) and EDS. A model X-ray diffractometer (XRD, “Bruker” D8 Advance, Karlsruhe, Germany) was used for observation; the copper target Kαray was used, the voltage was set at 40 kV, the current was 30 mA, the measurement angle (2θ) was 10°~90°, the step was 2θ = 0.02°, the crystal structure of the sample was observed and determined, and the phase was analyzed by Jade software.

A portable roughness tester, Handsurf (Accretech, Hitachi High-Tech, Tokyo, Japan), was used to measure the surface roughness and three-dimensional profile of the composite plating at different PTFE concentrations. The Vickers hardness of the plating was tested using a constant MH-5D Vickers hardness tester. To ensure the accuracy of the test results, each sample was measured three times and averaged.

### 2.3. Characterization of Coating Properties

The corrosion rate of the composite coatings was calculated by the weight loss method to characterize the corrosion resistance of the composite coatings. Q235B mild steel and three kinds of PTFE composite coatings with different concentrations were respectively placed in 5 wt% NaCl solution and immersed in seal corrosion for a week at room temperature. After the corrosion immersion, it was removed, cleaned, and blown dry to observe the macroscopic morphology and weighed to calculate the corrosion weight loss rate.

The electrochemical workstation CS350M was used to test the electrochemical impedance spectroscopy (EIS) and conduct corrosion tests. The test system was a three-electrode system consisting of a platinum electrode (counter electrode), KCl saturated calomel electrode (reference electrode), and composite coating sample (working electrode). To facilitate testing, composite coated samples were made to a suitable size. A copper wire was welded on the top of the composite coating sample to conduct electricity. In addition to leaving a suitable size of the test surface on the surface of the composite coating sample, the remaining area was sealed with epoxy resin. During the measurement, the sealed samples of Q235B mild steel and three kinds of composite coatings with different PTFE concentrations were immersed in 3.5 wt% NaCl solution, and the working electrode was immersed in the corrosive medium for a few minutes. After the open-circuit voltage was stable, the test work was carried out. The scanning voltage range of the polarization curve was 0.8~0.2 V, and the scanning rate was 1 mV/s. The scanning frequency of the ac impedance EIS was 10^−2^~10^5^ HZ. The corrosion current density (I_corr_) and corrosion potential (E_corr_) obtained after the electrochemical test met the Tafel equation [31]:E_corr_ = a + blogI_corr_

In the above formula, a and b are constants.

## 3. Results Discussion

### 3.1. Surface Morphology and Compositional Analysis of Ni-Cu-P-PTFE Composite Plating

PTFE is a polymer with strong chemical stability. It does not participate in any chemical reactions during the plating process; it only co-deposits on the surface of the substrate, with Ni, Cu, and P produced by the chemical reaction in the compound plating solution. In the composite plating solution, PTFE is adsorbed and transferred to the substrate surface by the catalytic dispersion of a cationic surfactant (CTAB) and the action of electrostatic field and then co-deposited in the composite plating layer, with Ni, Cu, and P elements produced by the chemical reaction, thus gradually forming a Ni-Cu-P-PTFE composite plating layer. Particle co-deposition is roughly divided into three major steps: PTFE particles are passed onto the substrate surface by convection; charged metal ions are weakly adsorbed on the cathode surface; and the reduced ions adsorb PTFE particles into the metal substrate. That is, H_2_PO_2_^−^ reaches the cathode surface to release electrons, and not only is Ni^2+^ reduced to the substrate surface accompanied by part of the current, but it also acts to reduce the surfactant and H^+^. The reaction process is as follows:H_2_PO_2_^−^ → HPO_2_^−^ + H
HPO_2_^−^ + OH^−^ → H_2_PO_3_^−^ + e
Cu^2+^ + 2e → Cu
H + H → H_2_
Total reaction equation: 2H_2_PO_2_^−^ + Cu^2+^ + 2OH^−^ → Cu + 2H_2_PO_3_^−^ + H_2_

Figure 2 shows the surface and section morphology of Ni-Cu-P-PTFE composite coatings with PTFE concentrations of 10 mL/L, 15 mL/L, and 20 mL/L, respectively. It can be observed in the SEM diagram that the composite coatings with different PTFE concentrations completely covered the surface of the matrix, in which the PTFE particles could be uniformly and tightly adsorbed on the composite coating. The microscopic morphology of the composite coatings was not different and was generally similar, and there were no obvious scratches, cracks, peeling, or other defects on the surface of the composite coatings. When the PTFE concentration was 10 mL/L, some large convex PTFE particles could be seen on the surface of the composite coating, and the surface was rough. Compared with the microstructure of the 10 mL/L PTFE concentration, when the PTFE concentration was 15 mL/L and 20 mL/L, the PTFE particles on the surface of the composite coating were obviously dispersed more evenly, and the surface was relatively smooth. Different concentrations of PTFE were uniformly dispersed and coated on the substrate surface by the dispersion of the surfactant CTAB, among which some PTFE particle agglomeration was observed on the surface of the composite coating with a PTFE concentration of 10 mL/L, and no obvious PTFE particle agglomeration was observed on the surface of the composite coating with PTFE concentrations of 15 mL/L or 20 mL/L. It can be seen in the microstructure of the composite coating section that PTFE particles were uniformly distributed in Ni-Cu-P-PTFE composite coating under the action of CTAB dispersion. The coating thickness was uniform and tightly bonded with the matrix, there were no microscopic cracks or bubbles, and the thickness of the three composite coatings was about 15 μm.

Figure 3 shows the surface energy dispersive spectrometer (EDS) element distribution of Ni-Cu-P-PTFE composite plating with PTFE concentrations of 10 m L/L, 15 mL/L, and 20 mL/L, respectively. As can be seen in the EDS scanning energy spectrum, nickel (Ni), copper (Cu), phosphorus (P), fluorine (F), and the other four elements of the three coatings had no obvious agglomeration phenomenon, and the dispersion was relatively uniform. It also shows that under the dispersive action of the surfactant (CTAB) [32], PTFE particles were uniformly suspended in the bath, and co-deposition occurred between particles so that PTFE particles did not show agglomeration in the bath. Using a constant temperature magnetic stirrer and a surfactant (CTAB), the PTFE particles formed a uniform suspension system in the bath [25]. Under the action of constant stirring and electrostatic charge adsorption, PTFE particles were encapsulated by Ni, Cu, P, and other elements, and co-deposition formed on the surface of the matrix. Due to the difference in the amount of PTFE emulsion and surfactant in the bath, the deposition amount of the PTFE particles on the composite coating had a certain difference. The EDS results showed that the other elements in the composite coating were more evenly distributed on the surface of the substrate. It can be seen that different concentrations of PTFE had no effect on the distribution of Ni, P, Cu, or F. According to the proportion of each element of the Ni-Cu-P-PTFE composite coatings in Table 2, PTFE particles were co-deposited and adsorbed on the matrix surface to become a component of the composite coating, and the distribution in the composite coating was relatively uniform. The main element of Q235B mild steel is Fe, and the presence of Fe was not found in the EDS surface energy spectrum of the Ni-Cu-P-PTFE composite coating, which indicates that the Ni-Cu-P-P-PTFE composite coating was successfully adsorbed on the surface of Q235B mild steel by chemical deposition.

### 3.2. Physical Phase Analysis of Ni-Cu-P-PTFE Composite Plating

Figure 4 shows the XRD diffraction patterns of Ni-Cu-P-PTFE composite plating with Q235B mild steel and PTFE concentrations of 10 mL/L, 15 mL/L, and 20 mL/L. As can be seen from the XRD diffraction pattern, the XRD diffraction patterns of composite coatings prepared under different PTFE concentrations were very similar, and there was no obvious deviation between the left and right of the XRD patterns for different PTFE concentrations. The characteristic diffraction peak of the Ni (111) crystal plane with a broad bun appeared in the three coatings around 2θ = 45° [24,33]. It mainly existed in the Ni phase, and no obvious characteristic diffraction peaks of Cu, P, or F were detected, which indicated that Cu, P, and F were coated in face-centered cubic nickel lattice during the Ni-Cu-P-PTFE composite coating deposition. These results indicate that the characteristic diffraction peak corresponding to the Ni (111) crystal plane in Ni-Cu-P-PTFE composite coating is not affected by different concentrations of PTFE. The intensity of the characteristic diffraction front of the Ni (111) crystal plane first increased and then decreased. When the PTFE concentration was 15 mL/L, the characteristic diffraction peak of the Ni (111) crystal plane was the highest. It is generally believed that the P content in the coating determines the structure of the coating. When the P content in the coating is greater than 12%, the coating has an amorphous structure [34]. According to Table 2, the proportion of elements on the surface of the Ni-Cu-P-PTFE composite coating showed that with the increase in PTFE concentration, the P contents of three kinds of Ni-Cu-P-PTFE composite coatings with different PTFE concentrations were 31%, 27%, and 29%, respectively, and the P content shows a trend of decreasing first and then increasing, all of which were greater than 12%. Therefore, Ni-Cu-P-PTFE composite coating has an amorphous structure.

### 3.3. Surface Roughness of Ni-Cu-P-PTFE Composite Plating

Figure 5 shows the three-dimensional profile of Ni-Cu-P-PTFE composite coatings with PTFE concentrations of 10 mL/L, 15 mL/L, and 20 mL/L. It can be seen that when the PTFE concentration was 10 mL/L, the surface of the composite coating had an increased number and size of block red bulge areas, indicating that there were obvious PFTE particles on the surface of the composite coating in this area, and when the PTFE concentration was 15 mL/L or 20 mL/L, the surface of the composite coating block red area was relatively smaller and smoother. This shows that the PTFE particles on the surface of the composite coating in this area were more evenly dispersed and the surface was relatively smooth, which was consistent with the observation and comparison of SEM images on the surface of the Ni-Cu-P-PTFE composite coating with different PTFE concentrations in Figure 1. At the same time, the surfaces of Ni-Cu-P-PTFE composite coatings with different PTFE concentrations were scanned in a 3D profile, and the surface roughness values of the composite coatings with 10 mL/L, 15 mL/L, and 20 mL/L were 5.235, 1.547, and 1.18, respectively.

### 3.4. Microhardness of Ni-Cu-P-PTFE Composite Coatings

Figure 6 is a graph of the measured data results of Vickers hardness of Ni-Cu-P-PTFE composite plating for Q235B mild steel and PTFE concentrations of 10 mL/L, 15 mL/L, and 20 mL/L. It can be seen from the graph that the chemical deposition of Ni-Cu-P-PTFE composite plating on the surface of Q235B mild steel could increase the Vickers hardness of the Q235B mild steel surface. With the increase in PTFE concentration, the Vickers hardness of Ni-Cu-P-PTFE composite coating decreased first and then increased, and the Vickers hardness values were 654 HV, 515 HV, and 674 HV, respectively. According to the proportions of each element in the Ni-Cu-P-PTFE composite coating with different PTFE concentrations in Table 2 and the Vickers hardness value of the Ni-Cu-P-PTFE composite coatings with different PTFE concentrations in Figure 6, it can be seen that the higher the content of the F element in the Ni-Cu-P-PTFE composite coating, the smaller the Vickers hardness of the Ni-Cu-P-PTFE composite coating; that is, the content of PTFE in the Ni-Cu-P-PTFE composite coating is inversely proportional to the Vickers hardness value of the Ni-Cu-P-PTFE composite coating. When the content of the F element (PTFE) in the Ni-Cu-P-PTFE composite coating was 31%, the Vickers hardness of the composite coating reached the minimum value of 515 HV.

Ni-Cu-P-PTFE composite plating on the surface of Q235B mild steel increased the Vickers hardness of the Q235B mild steel surface. However, the Vickers hardness of Ni-Cu-P-PTFE composite plating with different concentrations of PTFE is different; the Vickers hardness of Ni-Cu-P-PTFE composite plating with a PTFE content of 15 mL/L is the smallest, which was caused by the physical properties of PTFE itself, which has a soft and inelastic structure, so the elastic modulus of the alloy plating is much larger than that of PTFE [35]. The results showed that Ni-Cu-P-PTFE composite plating can easily produce plastic deformation due to the influence of external loading, thus reducing the Vickers hardness of the composite plating. PTFE has a particle dispersion enhancement effect in the Ni-Cu-P-PTFE composite plating, but the enhancement effect cannot be equivalent to the softening of PTFE, so the higher the PTFE content in the Ni-Cu-P-PTFE composite plating, the lower the Vickers hardness of the Ni-Cu-P-PTFE composite layer. In addition, the addition of PTFE reduced the effective bearing area of Ni-Cu-P-PTFE composite plating and increased the loss of Ni, Cu, and P particles in the Ni-Cu-P-PTFE composite plating, which also caused a decrease in the hardness of the Ni-Cu-P-PTFE composite plating.

### 3.5. Analysis of the Corrosion Performance of Ni-Cu-P-PTFE Composite Coatings

#### 3.5.1. Salt Corrosion Weight Loss Experiments

Figure 7 shows the macroscopic surface morphology of Q235B mild steel and the Ni-Cu-P-PTFE composite coating with PTFE concentrations of 10 mL/L, 15 mL/L, and 20 mL/L after 7 days of corrosion in 5 wt% NaCl solution. The surface of Q235B mild steel showed obvious corrosion pits, the specimens with 15 mL/L and 20 mL/L PTFE showed some local corrosion pits, and the specimens with 10 mL/L PTFE did not show any corrosion pits.

Figure 8 shows the corrosion rate of Q235B mild steel and Ni-Cu-P-PTFE composite coating with different PTFE concentrations in 5 wt% NaCl solution. It can be seen in the figure that the corrosion rate of the Q235B mild steel sample was the highest, and the corrosion rate of the Ni-Cu-P-PTFE composite coating sample with PTFE concentrations of 10 mL/L, 15 mL/L, and 20 mL/L increased successively. It can be seen that with the addition of PTFE, the corrosion resistance of Ni-Cu-P-PTFE composite plating was better than that of Q235B mild steel. Among them, the Ni-Cu-P-PTFE composite coating with a PTFE concentration of 10 mL/L had the best corrosion resistance.

#### 3.5.2. Electrochemical Impedance Spectroscopy Analysis

Figure 9 shows the electrochemical impedance spectroscopy (EIS) data of Q235B mild steel and Ni-Cu-P-PTFE composite coatings with three different PTFE concentrations in 3.5 wt% NaCl solution. It can be seen from the results of the impedance diagram that the impedance arc diameters of Ni-Cu-P-PTFE composite coatings with three different concentrations of PTFE were larger than those of Q235B mild steel. With the increase in PTFE concentration, the impedance arc diameters of the Ni-Cu-P-PTFE composite coatings showed a gradually decreasing trend. Generally, the larger the diameter of the impedance arc, the better the corrosion resistance of the material [36]. Obviously, the corrosion resistance of Ni-Cu-P-PTFE composite coatings with three different PTFE concentrations is better than that of Q235B mild steel. These results were consistent with the results of the weightlessness experiment.

In order to better understand the impedance characteristics of Q235B mild steel and three different PTFE concentrations of Ni-Cu-P-PTFE composite plating in 3.5 wt% NaCl solution for corrosion protection, the impedance spectra of Q235B mild steel and three different PTFE concentrations of Ni-Cu-P-PTFE composite plating specimens were fitted using Zview software to obtain the best equivalent circuit, as shown in Figure 10. The impedance values of the circuit elements fitted with Q235B mild steel and three different PTFE concentrations of Ni-Cu-P-PTFE composite plating are shown in Table 3, where R_s_ is the resistance of 3.5 wt% NaCl solution; CPE_dl_ is the constant phase angle element between the electrode surface film and the electrode; R_ct_ represents the charge transfer resistance of the metal corrosion reaction; and Z_w_ is the Warburg impedance.

The fitting results showed that the fitting errors of all parameters were within 10%, which indicates that the fitting effect was good, and the equivalent circuit matched the experimental data well. As can be seen in Table 3, the R_ct_ values of the Ni-Cu-P-PTFE composite plating with three different PTFE concentrations were much larger than those of Q235B mild steel, indicating that the corrosion resistance of the Ni-Cu-P-PTFE composite plating is better than that of Q235B mild steel. The R_ct_ value of the Ni-Cu-P-PTFE composite coating with a PTFE concentration of 10 mL/L was the largest, so the Ni-Cu-P-PTFE composite coating with 10 mL/L had the best corrosion resistance among the three different PTFE concentrations of Ni-Cu-P-PTFE composite coating.

#### 3.5.3. Dynamic Potential Polarization Curve Analysis

Figure 11 shows the polarization curves of Q235B mild steel and three different PTFE concentrations of Ni-Cu-P-PTFE composite coatings in 3.5 wt% NaCl solution, and the electrochemical corrosion data fitted by software calculations are shown in Table 4. Combined with the analysis in Figure 10 and Table 3, it can be seen that the corrosion current densities (I_corr_) of the three different PTFE concentrations of Ni-Cu-P-PTFE composite coatings were all lower than those of Q235B mild steel, and the corrosion potentials (E_corr_) were all more positive than those of Q235B mild steel. The Q235B mild steel exhibited the lowest Ecorr, with a value of −0.805 V, and Icorr, with a value of 8.499 × 10^−5^ A·cm^−2^, compared with the three Ni-Cu-P-PTFE composite coatings. For the three Ni Cu-P-PTFE composite coatings, Ecorr became higher and Icorr became lower.

Generally speaking, the lower the corrosion current density and the more positive the corrosion potential, the stronger the corrosion resistance of the material [37]. It can be seen that the corrosion resistance of the Ni-Cu-P-PTFE composite coating is better than that of Q235B mild steel. The best corrosion resistance was achieved at a PTFE concentration of 10 mL/L, which was consistent with the results of the previous weight loss experiments and EIS analysis. Due to the high hydrophobicity and excellent chemical stability of PTFE particles, water molecules containing corrosive media do not easily penetrate the Ni-Cu-P-PTFE composite coating and enter the substrate, thus improving the corrosion resistance of the Ni-Cu-P-PTFE composite layer. However, when the concentration of PTFE in the plating solution is high due to the limited dispersion effect of the added quantitative surface active agent CTAB, the dispersion effect of the surface active agent CTAB is weakened, making the spacing between the PTFE particles adsorbed on the surface of the substrate larger, which leads to more defects; thus, the corrosion medium invades from the defects and comes into contact with the Q235B mild steel phase, resulting in a decrease in the corrosion resistance of the Ni-Cu-P-PTFE composite coating [38].

## 4. Conclusions

In this paper, Ni-Cu-P-PTFE composite coating was prepared on Q235B mild steel by the chemical composite plating method, and the corrosion resistance of the coating to high Cl^−^ ions was explored. The following conclusions were obtained:
Different concentrations of PTFE have almost no effect on the physical phase structure of Ni-Cu-P-PTFE composite plating; Ni-Cu-P-PTFE composite plating has an amorphous structure, the microscopic morphology of Ni-Cu-P-PTFE composite plating surface and cross-section is smooth and dense, and PTFE can be uniformly co-deposited on the substrate surface under the dispersion of CTAB. The three-dimensional contour results showed that the surface roughness of the composite plated layer with a PTFE concentration of 10 mL/L was larger, with a surface roughness value of 5.235. The surface roughness values of the PTFE concentrations of 15 mL/L and 20 mL/L were smaller, with surface roughness values of 1.547 and 1.18, respectively.The Ni-Cu-P-PTFE composite coating on the surface of Q235B mild steel can increase the Vickers hardness of its surface. Due to the soft physical properties and loose structure of PTFE, the Vickers hardness of Ni-Cu-P-PTFE composite coating with different concentrations of PTFE varies; the Vickers hardness of Ni-Cu-P-P-PTFE composite coating reached a minimum value of 515 HV when the PTFE concentration was 15 mL/L.The results of both the weight loss experiments and electrochemical tests showed that the corrosion resistance of Ni-Cu-P-PTFE composite coating was better than that of Q235B mild steel. The best corrosion resistance was the Ni-Cu-P-PTFE composite coating with a PTFE concentration of 10 mL/L. The weight loss rate was 0.24 mg∙cm^−2^, the corrosion current density was 7.255 × 10^−6^ A∙cm^−2^, and the corrosion voltage was −0.314 V. In addition, the corrosion resistance of the Ni-Cu-P-PTFE composite plating became worse with the increase in PTFE concentration.There are few studies on the preparation of Ni-Cu-P-PTFE composite coatings and their properties, and there are almost no studies on the influence of the introduction of PTFE particles on the surface morphology, organization, corrosion resistance, and mechanical properties of Ni-Cu-P-PTFE composite coatings. This also brings difficulties and uncertainties to the application of Ni-Cu-P-PTFE composite coatings. In this paper, three Ni-Cu-P-PTFE composite coatings with different PTFE concentrations were tested and characterized in terms of surface morphology, organization, corrosion resistance, and mechanical properties, which is useful for guiding the performance of Ni-Cu-P-PTFE composite coatings and the deposition mechanism.


## Figures and Tables

**Figure 1 materials-16-01966-f001:**
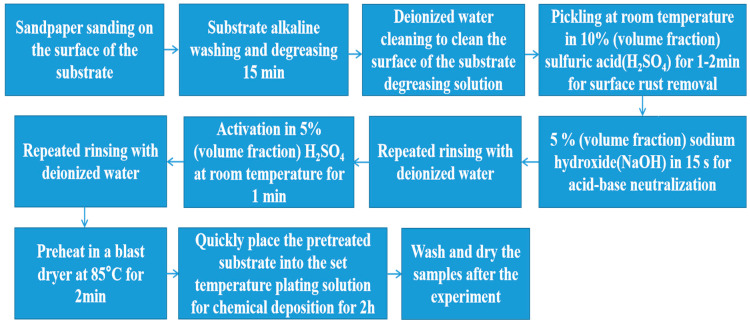
Flow chart of experimental process setup.

**Figure 2 materials-16-01966-f002:**
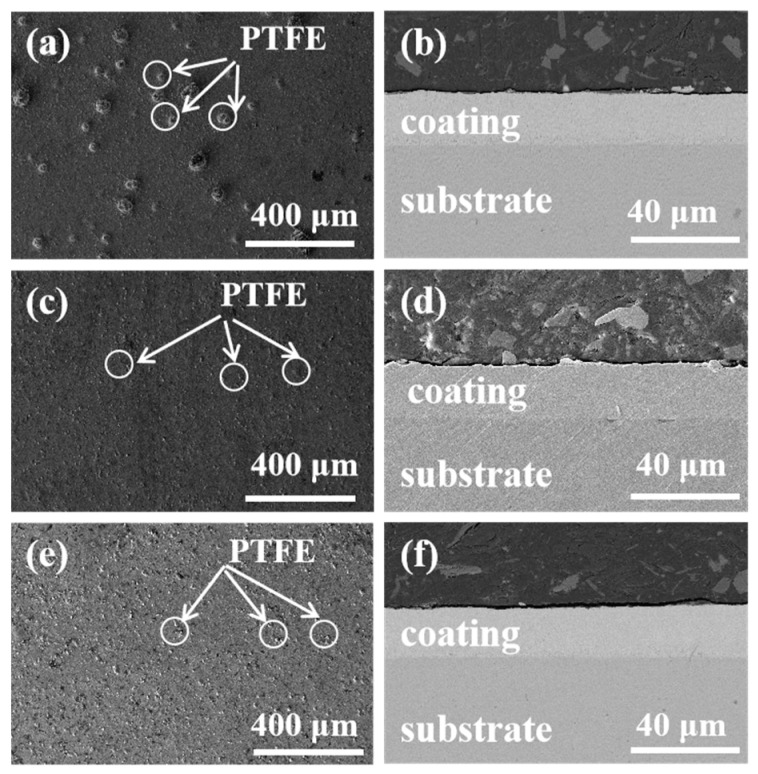
SEM images of surface and cross-section of Ni-Cu-P-PTFE composite coating: PTFE (10 mL/L) (**a**,**b**); PTFE (15 mL/L) (**c**,**d**); PTFE (20 mL/L) (**e**,**f**).

**Figure 3 materials-16-01966-f003:**
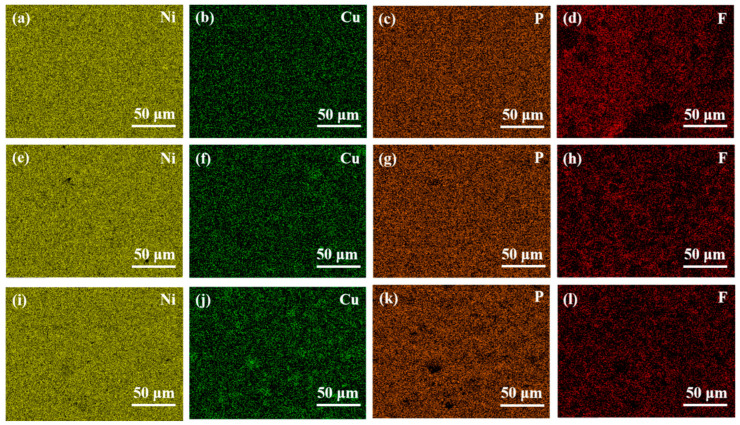
Surface energy spectrum of Ni-Cu-P-PTFE composite plating: PTFE (10 mL/L) (**a**–**d**); PTFE (15 mL/L) (**e**–**h**); PTFE (20 mL/L) (**i**–**l**).

**Figure 4 materials-16-01966-f004:**
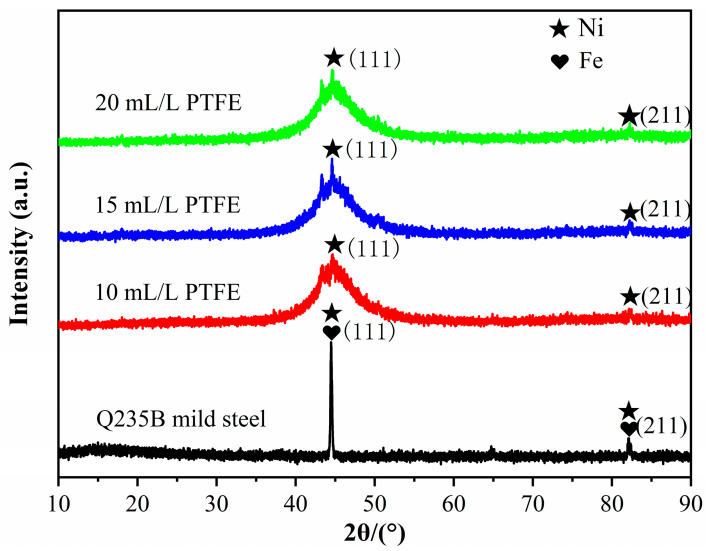
XRD patterns of Ni-Cu-P-PTFE composite plating of Q235B mild steel and different PTFE concentrations.

**Figure 5 materials-16-01966-f005:**
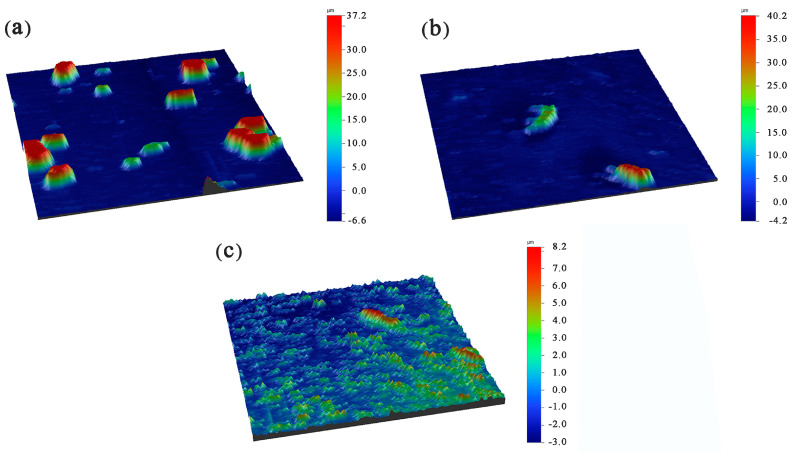
Three-dimensional surface profile of Ni-Cu-P-PTFE composite coating: PTFE (10 mL/L) (**a**); PTFE (15 mL/L) (**b**); PTFE (20 mL/L) (**c**).

**Figure 6 materials-16-01966-f006:**
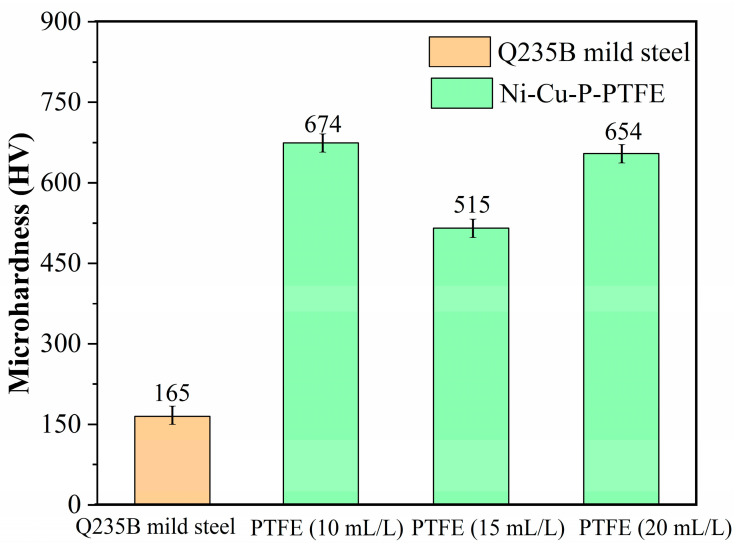
Vickers hardness of Ni-Cu-P-PTFE composite plating on Q235B mild steel and different PTFE concentrations.

**Figure 7 materials-16-01966-f007:**
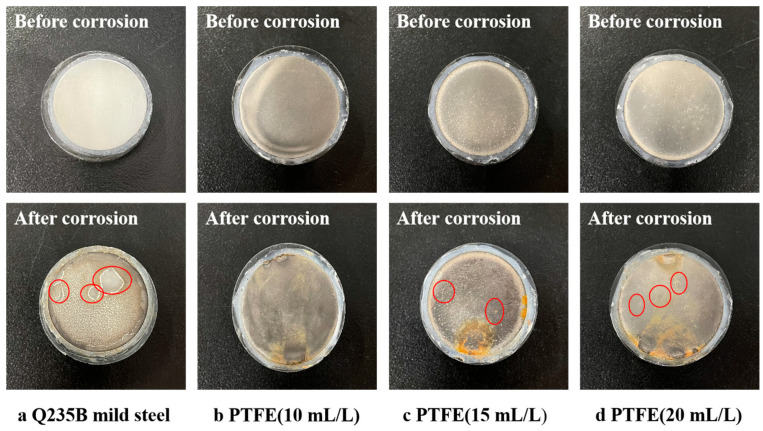
The surface macromorphology of Q235B mild steel and Ni-Cu-P-PTFE composite coatings with different PTFE concentrations before and after corrosion. The red circled ones represent corrosion pits.

**Figure 8 materials-16-01966-f008:**
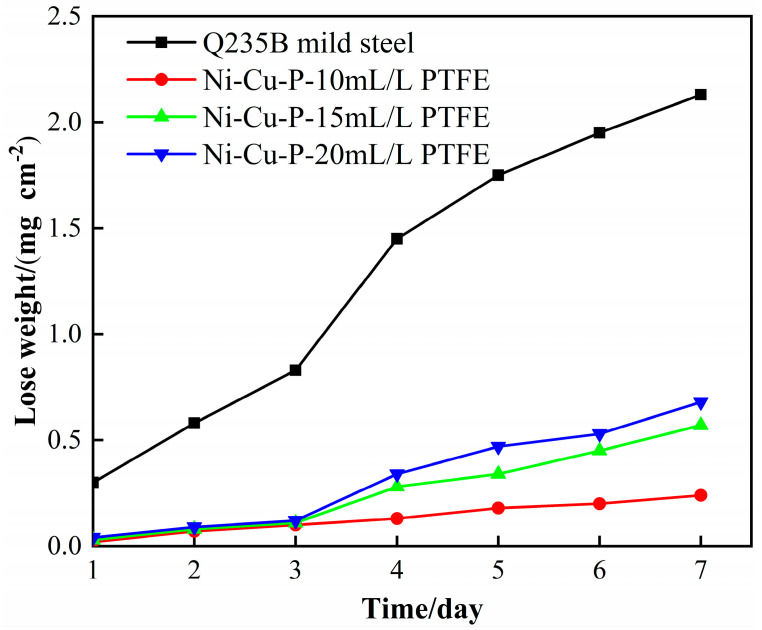
The corrosion rate of Q235B mild steel and Ni-Cu-P-PTFE composite coatings with different PTFE concentrations in 5 wt% NaCl solution.

**Figure 9 materials-16-01966-f009:**
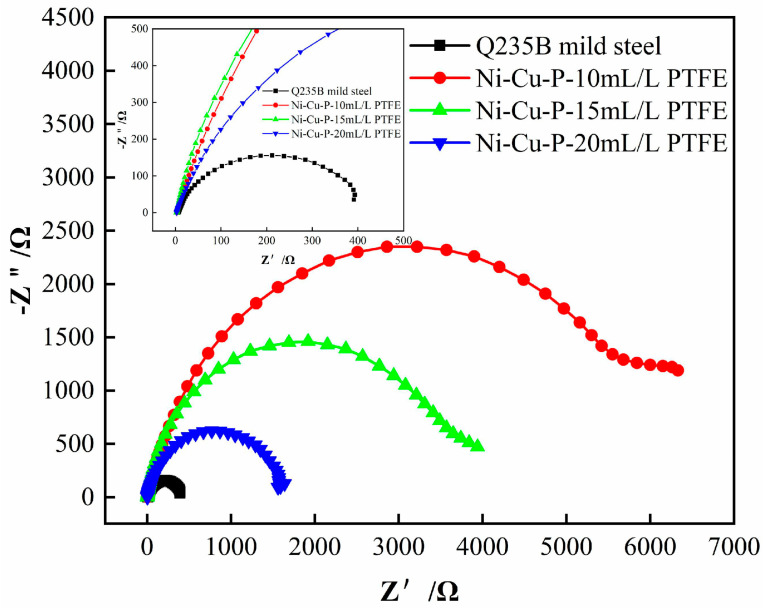
Impedance diagram of Ni-Cu-P-PTFE composite coating for Q235B mild steel and different PTFE concentrations.

**Figure 10 materials-16-01966-f010:**
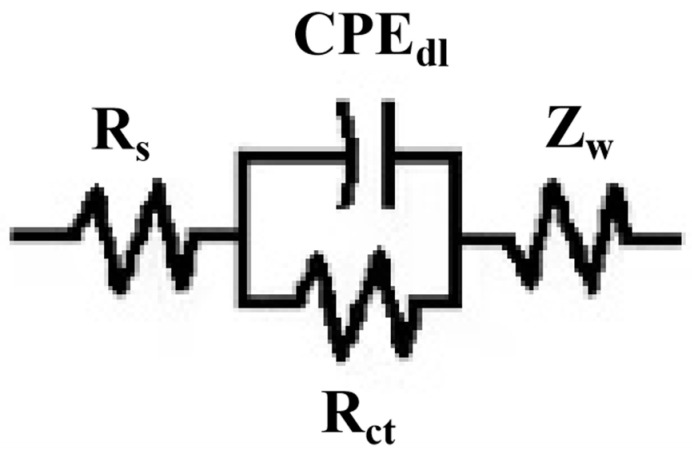
Best-fit circuit diagram for Ni-Cu-P-PTFE composite plating of Q235B mild steel and different PTFE concentrations.

**Figure 11 materials-16-01966-f011:**
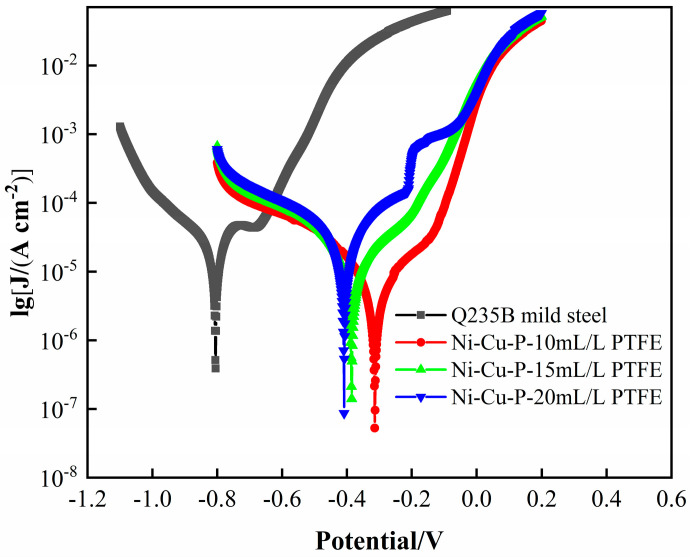
Tafel curves of Q235B mild steel and Ni-Cu-P-PTFE composite coatings with different PTFE concentrations in 3.5 wt% NaCl solution.

**Table 1 materials-16-01966-t001:** Composition and technological conditions of Ni-Cu-P-PTFE plating solution.

Reagent Name	Sample 1	Sample 2	Sample 3
NiSO_4_∙6H_2_O	30 g/L	30 g/L	30 g/L
CuSO_4_∙5H_2_O	0.8 g/L	0.8 g/L	0.8 g/L
NaH_2_PO_2_∙H_2_O	30 g/L	30 g/L	30 g/L
C_6_H_5_Na_3_O_7_∙2H_2_O	6 g/L	6 g/L	6 g/L
C_3_H_6_O_3_	15 mL/L	15 mL/L	15 mL/L
CH_3_COONa	15 g/L	15 g/L	15 g/L
C_2_H_5_NO_2_	5 g/L	5 g/L	5 g/L
CH_4_N_2_S	suitable amount	suitable amount	suitable amount
CTAB	0.5 g/L	0.5 g/L	0.5 g/L
60 wt% PTFE	10 mL/L	15 mL/L	20 mL/L
Temperature	85 °C~87 °C	85 °C~87 °C	85 °C~87 °C
PH	4.5~4.6	4.5~4.6	4.5~4.6

**Table 2 materials-16-01966-t002:** Percentage of each element on the surface of Ni-Cu-P-PTFE composite coating with different PTFE concentrations.

Ni-Cu-P-PTFE			
	PTFE (10 mL/L)	PTFE (15 mL/L)	PTFE (20 mL/L)
Element	Weight %	Weight %	Weight %
F	26	31	28
P	31	27	29
Ni	34	36	35
Cu	9	6	8

**Table 3 materials-16-01966-t003:** Impedance values of the best-fit circuit for Q235B mild steel and Ni-Cu-P-PTFE composite plating with different PTFE concentrations.

Simple	R_s_/Ω·cm^2^	R_ct_/Ω·cm^2^
Q235B mild Stell	1.014	387
Ni-Cu-P-10 mL/L PTFE	2.987	5.386 × 10^3^
Ni-Cu-P-15 mL/L PTFE	2.654	3.875 × 10^3^
Ni-Cu-P-20 mL/L PTFE	2.446	1.683 × 10^3^

**Table 4 materials-16-01966-t004:** Electrochemical parameters of different samples in 3.5 wt% NaCl solution.

Simple	E_corr_ /V	I_corr_ /(A∙cm^−2^)	OCP/V
Q235B mild Stell	−0.805	8.499 × 10^−5^	−0.549
Ni-Cu-P-10 mL/L PTFE	−0.314	7.255 × 10^−6^	−0.307
Ni-Cu-P-15 mL/L PTFE	−0.385	1.551 × 10^−5^	−0.338
Ni-Cu-P-20 mL/L PTFE	−0.409	2.508 × 10^−5^	−0.377

## Data Availability

Not applicable.

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
