# Peer review of "Effect of Polytetrafluoroethylene (PTFE) Content on the Properties of Ni-Cu-P-PTFE Composite Coatings"

_materials, 2023, doi:10.3390/ma16051966_

Round 1
Reviewer 1 Report
The manuscript by X. Liang et al. presents a study of the polytetrafluoroethylene-coated steel by means of microstructure, elemental and phase composition, hardness and corrosion test. The most interesting result reveals the corrosion protection after coating. However, the manuscript should be improved before publication.
1) All abbreviations should be described before they appear in the text. For example, PTFE in the title, and many other examples throughout the manuscript.
2) English grammar and spelling should be carefully checked, especially in the Abstract and Methods sections.
3) Table 1: it looks like the second sample has all ingredients, while 1st and 3rd - only PTFE. Please, improve the formatting of Table 1.
4) The XRD machine should be "Bruker" D8 Advance. Its normal operation is 40kV and 40mA, which gives 1.6kW. The authors used 40kV and 100mA (4000 W). Is it safe for X-ray tube? Please, check.
5) The phase composition: Figure 3 shows a broad peak around ~45 degrees. It can be attributed to iron steel. In my opinion, Figure 3 shows XRD of substrate rather than coating.
6) The hardness test: the results of X-ray diffraction as well as the results of hardness test of the coated materials, which are presented in Figures 3 and 5, respectively, should be compared with the iron steel without coating.
Author Response
Hello external reviewer, please see the attachment for the revision.

Reviewer 2 Report
This manuscript considers the effect of polytetrafluoroethylene (PTFE) on some properties of Ni-Cu-P/PTFE composite coatings electrodeposited from an aqueous sulfate electrolyte.
The manuscript contains a number of new data that may be of interest and useful to a wide range of the readers of this Journal. In my opinion, the manuscript could be recommended for publication after taking into account a number of the following comments.
1. It is well known that it is advisable not to use abbreviations in the titles of articles, abstracts and lists of keywords. Perhaps the authors will be able to update the manuscript taking into account this circumstance. At least the abbreviation PTFE should be explained at the first mention in the abstract.
2. When discussing the deposition of composite coatings in introduction section, it would be strange not to mention key reviews on this topic (see, for example, doi: 10.1016/j.coelec.2020.01.01, doi: 10.1080/00202967.2020.1819022, and doi: 10.1179/0020296713Z.000000000161). In addition, the authors should draw attention to the following useful publications: Electrodeposition of nanostructured nickel-ceramic composite coatings: a review, Int. J. Electrochem. Sci., 9 (2014) 1942–1963; and Electrochemical preparation and characterization of Ni–PTFE composite coatings from a non-aqueous solution without additives, Int. J. Electrochem. Sci., 7 (2012) 12440–12455.
3. It is necessary to substantiate the choice of the electrolyte composition, concentration of components and the alloy deposition conditions (pH and temperature). Are these parameters optimal?
4. Although the authors concentrate all their efforts on studying the properties of coatings, and such an approach can be justified, nevertheless, at the beginning of the main part of the manuscript (Results and discussion), at least a few sentences or paragraphs devoted to the features of the coating synthesis should be given. It would be necessary to indicate the possible mechanism of the processes and give the equations of chemical reactions leading to the formation of coatings. This would contribute to a smoother and more consistent presentation of the manuscript results.
5. Discussing a possible mechanism for the inclusion of PTFE particles in a metal matrix and mentioning the adsorption of these particles, one should cite recent publications that consider this issue (see, for example, doi: 10.1016/j.jelechem.2022.116463) and briefly discuss the kinetics of the process in the context of developed theoretical concepts.
6. Data on the chemical composition of coatings (Table 2) should be supplemented with information on the chemical composition of coatings without PTFE particles (for a "pure" alloy, not a composite). Accordingly, it is necessary to expand the discussion.
Moreover, in my opinion, this remark concerns not only the data on the chemical composition, but also on all the studied properties and microstructure of the coatings: the data obtained for composites with different PTFE content must necessarily be compared with the corresponding data for a "pure" alloy (without PTFE). This is one of the main and key remarks to the manuscript.
7. By the way, Table 2 is built incorrectly. It should contain only one column listing the names of chemical elements. There is no need to duplicate it three times.
8. The authors used the method of electrochemical impedance spectroscopy to study the corrosion behavior. However, this method is not a qualitative, but a quantitative research method, i.e. the obtained data should be quantified (choose the proper equivalent circuit and calculate the numerical characteristics of the individual elements of this circuit). By the way, the resulting Nyquist diagrams are not one semicircle, but an overlay of two semicircles. What features of the mechanism and kinetics of the corrosion process can be associated with such behavior?
9. The abstract to the manuscript includes a large number of actual experimental data, but does not contain elements of discussion and scientific generalizations. The abstract should be rewritten properly.
10. It is necessary to significantly strengthen the elements of discussion and generalization of the obtained data. The manuscript includes a large amount of experimental data, but their discussion is superficial. Proper conclusions and generalizations are not made in full. This deficiency should be corrected.
11. The name of the same manufacturer of the used reagents is obsessively repeated in the Experimental section twelve times (!). Please arrange these data in such a way that you do not repeat the same information many times, which is not of paramount importance for a scientific message.
12. Numerous typos in the text should be corrected (for example, in Table 3 "Ecorr" and "Icorr" should be instead of "Ecoor" and "Icoor", respectively; on page 4 (line 152) "NaCl solution" should be instead of "Nacl solution"; etc.).
Generally speaking, the English language of the manuscript is quite unsatisfactory. There are numerous grammatical errors. For example, there is no predicate in the first sentence of the abstract.
Author Response

(The authors gave the same response as above.)

Reviewer 3 Report
The manuscript investigates the effect of PTFE content on the properties of Ni-Cu-P-PTFE electroless coating. The experiments are predictable and superficial, several statements are contradictory to each other, and so a significant improvement is needed for this research to be published in a highly cited journal. Some comments are as follows:
Line 75: “..than the uncoated composite coating on soft steel surface.” What does it mean “the uncoated composite coating?”
Lines 117 – 132: improve language
Line 178, the PTFE particles increase the surface roughness. In many applications, high coating roughness in unacceptable. How are the authors going to solve this problem?
Line 182, authors claim that PTFE may be dispersed and coated on the surface of the matrix under the dispersive action of surfactant (CTAB), and no obvious agglomeration occurs. However, Figure 1 clearly shows agglomerates of PTFE. Explain this discrepancy.
Line 242, 3.3. Surface roughness of Ni-Cu-P-PTFE composite plating: how is it possible that coatings with higher PTFE content are smoother than the coating with lower PTFE content? Why surfactant action was not beneficial in case when 10 ml/l PTFE in solution was used?
Since the paper goal is to get insight into the PTFE effect on the coating characteristics, it is obligatory to provide results of hardness and corrosion for PTFE-free samples (Ni-Cu-P coating). Maybe PTFE-free samples show better results?
In line 300, authors state that “…the corrosion rate of composite coating with the PTFE concentration of 10mL/L, 15 mL/L and 20 mL/L increases successively.”; but in line 304, they say that “..the corrosion resistance of Ni-Cu-P-PTFE composite coatings decreases with the increase of PTFE particle content.” These statements are contradictory.
Line 301: “It can be seen that the addition of PTFE particles makes the Ni-Cu-P-PTFE composite coating have certain corrosion resistance.” What does it mean? Every material has certain corrosion resistance.
Figure 7: explain better how you obtained the results. Was one sample immersed and measured every day, or more samples?
Line 342: authors claim that “…when the PTFE concentration is high, the dispersion effect is not very good… which leads to an increase in defects…” However, in line 179, authors claim that “…when the PTFE concentration is 15 mL/L and 20 mL/L, the PTFE particles on the surface of the composite coating are obviously dispersed more evenly and the surface is relatively smooth..”. Explain this inconsistency.
Corrosion experiments are superficial. Please provide literature data on corrosion of Ni-alloy electroless coatings, and give comparison with your results.
Author Response

(The authors gave the same response as above.)

Reviewer 4 Report
The authors studied the effects of different concentrations of PTFE on the physical phase composition, microscopic morphology and its properties of Ni-Cu-P-PTFE composite coatings. The manuscript had an interesting topic and was well-written, however, it could only be accepted with the following minor revisions:
1. Abstract: The reader may be confused by the brief terms PTFE, Ni, Cu, AC, SEM, EDS, NaCl, and XRD.
2. Lack of introduction and problem statement in the abstract.
3. Avoid from using acronyms (PTFE) in keywords.
4. Too many acronyms throughout the paper. It is suggested to create a list of abbreviations at the beginning of the paper.
5. Too many repetitive words, for example, “MACKLIN, Shanghai, China”, “Ni-Cu-P-PTFE”. Please improve to make your writing look nicer.
6. Refer to lines 103 – 114, what is this? It is difficult to read. The paragraph looks so weird and needs proofreading.
7. It is recommended that the authors provide a flow chart for the experimental setup.
8. In Figure 4, the scale bars were too tidy that cannot see.
9. Section 3.2, 3.3 and 3.4 there were no reflections of comparison with previous studies.
10. Table 1, why so many blanks in sample 1 and sample 2 columns?
11. Figure 5, the best of presenting the bar graph is by adding the SD (error bar).
12. It is suggested to add the limitation, future study and develop implications for researchers in the conclusion section.
Author Response

(The authors gave the same response as above.)

Round 2
Reviewer 1 Report
The authors correctly addressed all queries. The manuscript can be published in the present form.
Author Response
Thank you very much for your review of this article
Reviewer 2 Report
After reviewing the revised version of the manuscript, I state that the authors managed to somewhat improve their work, based on the suggestions and comments of the reviewers.
At the same time, I am extremely surprised by the fact that the authors completely ignored several of my comments (numbers 4, 5, 6, 8 and 12 in my initial review). Moreover, the authors in their responses to remark number 4 note the following:
"Response 4: Thanks for pointing out this issue. We have ... added the possible mechanisms that point to the process and give the chemical reaction equations that lead to the formation of the coating."
However, there is no such discussion in the revised version of the manuscript; the equations of chemical reactions are not given either.
Further, in response to remark number 5, the authors indicate the following:
"We have briefly discussed the kinetics of the process and made relevant literature references". At the same time, there are absolutely no corresponding changes and corrections in the manuscript.
With regard to remark number 6, then, again, although the authors declare that they provided the relevant experiments and data, these changes simply do not appear in the revised version of the manuscript.
Regarding remark number 8, in this case, too, the authors falsely answered that the corrections had been made, although in fact there was not even an attempt to carry out the corresponding modification.
Finally, the typos I pointed out in remark number 12 were also not corrected, despite the authors' claim that they were.
Thus, the authors knowingly informed the Editor of false information and completely ignored the reviewer's suggestions, while indicating in their answers that the corresponding modification of the manuscript had allegedly been carried out. This is the worst possible development of the situation.
Under the circumstances, I strongly insist that the authors listen to all my recommendations. The authors should give the equations of the reactions responsible for the formation of the material under study, briefly discuss the kinetics of the process of the composite formation, cite the mentioned publication (remarks numbered 4 and 5 ), present data on the composition and various properties of the alloy without PTFE particles (remark number 6), give an adequate quantitative interpretation of the electrochemical impedance spectroscopy data (remark number 8), and correct misprints (remark number 12). Only after such corrections, the manuscript can be recommended for publication.
Author Response
Thank you very much for your review of this article, the answers to the questions are attached

Reviewer 3 Report
The manuscript investigates the effect of PTFE content on the properties of Ni-Cu-P-PTFE electroless coating. Some parts of the manuscript have been improved in relation to the previous version, but there are still many weak spots that must be improved prior to publication, as follows:
Instead of Cl-1, write just Cl-.
“Herein, to explore the effects of different concentrations of Polytetrafluoroethylene(PTFE) on the physical phase composition, microscopic morphology and its properties of Ni-Cu-P-PTFE composite coatings.” erroneous grammar
Why is the first letter capital in chemical substance and element names?
“There are many preparation methods for corrosion resistant metal-particle composite coatings, such as electroplating [28-29], electrodeposition [30], chemical plating [31], etc. chemical plating is a simple and low-cost method.” erroneous grammar
“The specific electroless plating experiment process are as follows.” erroneous grammar
“First, the substrate first after 400#, 127 800#, 1200#, 1600#, 2000#, 2400# sandpaper step by step grinding, after grinding the surface of the substrate thrown into a bright mirror.” erroneous grammar
Page 3, lines 127-139: the sample preparation should not be described in imperative mode, but in past tense mode.
Page 5, line 189, “and the PTFE particles can be evenly and relatively flat and compact adsorbed on the composite coatings” erroneous grammar
Page 6, line 225: “EDS results show that the distribution of other elements in the composite coating on the substrate surface is relatively all.” Explain this sentence
Page 9, line 285: “With the increase of PTFE concentration, the Vickers hardness of Ni-Cu-P-PTFE composite coating decreases first and then increases, and the Vickers hardness values are 654 HV, 515 HV and 674 HV, respectively”. This text is not in correlation with figure 6.
Page 9, line 291: “the higher the content of F element in the Ni-Cu-P-PTFE composite coating, the higher the content of F element in the Ni-Cu-P-PTFE composite coating..” correct this
Page 9, line 304: “..but the enhancement effect cannot equalize the softening of PTFE, so the Ni-Cu-P-PTFE composite plating.” erroneous grammar
Page 9, line 298: “The addition of PTFE decreases the Vickers hardness of the Ni-Cu-P-PTFE composite plating…” How can one make this conclusion, without the hardness data for Ni-Cu-P (PTFE free) sample?
Figure 6: please provide hardness for Ni-Cu-P (PTFE free) sample.
Page 9, line 298: “and the Vickers hardness value of the Ni-Cu-P-PTFE composite coating with different PTFE concentrations in Figure 5..” Figure 5 or 6?
Section 3.4. is baffling and contradictory. Does PTFE increase or decrease the coating hardness?
“Figure 8 shows the corrosion rate of Ni-Cu-P-PTFE composite coatings of Q235B mild steel and PTFE concentrations of 10 mL/L, 15 mL/L and 20 mL/L in 5%wt NaCl solution for 7 days after corrosion.” Baffling sentence, erroneous grammar.
3.5. Analysis of the corrosion performance: the data related to the corrosion of Ni-Cu-P (PTFE free) sample are still missing. How can one conclude anything about the influence of PTFE, when there are no data to compare coatings with and without PTFE?
Caption of Figure 8 is baffling, with erroneous grammar.
Figure 9, figure 10: what was the immersion period prior to EIS measurement and Tafel test?
Page 12, line 358: “The Q235B mild steel exhibits the noblest Ecorr with a value of -0.805 V and Icorr with a value of 8.499×10-5 A·cm-2, compared with the three Ni-Cu-P-PTFE composite coatings. For the three Ni Cu-P-PTFE composite coatings, Ecorr becomes corrected and icorr becomes lower.” Explain terms “the noblest Ecorr” and “Ecorr becomes corrected”.
Conclusion: …”and the corrosion resistance of the coating to high Cl-1 ions was explored.” Explain term “high Cl- ions”.
Page 13, line 388: “.substrate surface.” ?
The sentence in conclusion, “The Vickers hardness is inversely proportional to the content of PTFE in the Ni-Cu-P-PTFE composite coating, i.e., the more PTFE in the Ni-Cu-P-PTFE composite coating, the lower the Vickers hardness of the Ni-Cu-P-PTFE composite coating.”, does not corelate with figure 6.
Conclusion, “…in which the best corrosion resistance was the Ni-Cu-P-PTFE composite coating with PTFE concentration of 10 mL/L.” erroneous grammar
Page 13, line 405: “The weight loss rate was 0.24 mg∙cm-2…” Please explain how you obtained this result.
Conclusion, “Due to the limitation of time and experimental conditions, part of the work has not been perfected…” this text is unnecessary.
Since the paper goal is to get insight into the PTFE effect on the coating characteristics, it is obligatory to provide results of hardness and corrosion for PTFE-free samples (Ni-Cu-P coating). Or at least provide data from earlier references. I do not see the data for Ni-Cu-P coating from earlier references.
Author Response

(The authors gave the same response as above.)

Reviewer 4 Report
All reviewing questions have been answered.
Author Response

(The authors gave the same response as above.)
